# Jennifer for COVID-19: An NLP-Powered Chatbot Built for the People and by the People to Combat Misinformation

**Yunyao Li**[1,*]   **Tyrone Grandison**[2,*]   **Patricia Silveyra**[3,*]   **Ali Douraghy**[4]
**Xinyu Guan**[5]   **Thomas Kieselbach**[6]   **Chengkai Li**[7]   **Haiqi Zhang**[7]

[1] IBM Research - Almaden [2] The Data-Driven Institute [3] University of North Carolina - Chapel Hill
[4] The National Academies of Sciences, Engineering and Medicine [5] Yale University
[6] Umeå University [7] University of Texas - Arlington

{yunyaoli@us.ibm.com, tgrandison@data-driven.institute, patry@email.unc.edu, adouraghy@nas.edu}

## Abstract

Just as SARS-CoV-2, a new form of coronavirus continues to infect a growing number of people around the world, harmful misinformation about the outbreak also continues to spread. With the goal of combating misinformation, we designed and built `Jennifer`–a chatbot maintained by a global group of volunteers. With `Jennifer`, we hope to learn whether public information from reputable sources could be more effectively organized and shared in the wake of a crisis as well as to understand issues that the public were most immediately curious about. In this paper, we introduce `Jennifer` and describe the design of this proof-of-principle system. We also present lessons learned and discuss open challenges. Finally, to facilitate future research, we release COQB-19 (COVID-19 Question Bank)[1], a dataset of 3,924 COVID-19-related questions in 944 groups, gathered from our users and volunteers. `Jennifer` is available at http://bit.ly/jenniferai.[2]

## 1 Introduction

This paper introduces `Jennifer`, a chatbot created to provide easily accessible information from reliable resources to answer questions related to the current COVID-19 pandemic. `Jennifer` leverages cutting-edge chatbot technology, as well as a global network of volunteers, to combat misinformation related to the pandemic. The information provided by `Jennifer` covers a wide variety of topics, ranging from case statistics and food safety to best practices for disease prevention and management.

The idea of `Jennifer` was born in early March 2020 during the semi-annual meeting of New Voices,[3] a project of the National Academies of Sciences, Engineering, and Medicine.[4] The New Voices members, a group of early-career scientists representing a diversity of research, health and policy perspectives, knew that the scientific community could rapidly mobilize its expertise to address the public health challenges the United States would soon face, and called for "*rapid collaborations between scientists and the civic tech communities to educate the public*" (New Voices, 2020). We envisioned using the latest techniques in artificial intelligence to create a platform of evidence-based information from reliable sources that the public would find easy to interact with.

We quickly mobilized to design and build `Jennifer` and to recruit a global group of volunteer scientists to help test and scale `Jennifer`'s performance. The proof-of-principle system will demonstrate the feasibility to directly crowd-source the global scientific community's expertise for public benefit without the need for intermediaries, and help improve public trust in science.

Our core design considerations are:

- **Rapid Development**: `Jennifer` should be built within a short amount of time to win the race against fast spreading misinformation.
- **Ease of Access**: `Jennifer` should provide information to the general public in an easily accessible manner across different platforms (e.g. Web and social media).
- **Ease of Maintenance**: `Jennifer` should be maintainable by broader and more diverse group of volunteers.
- **Quality Assurance**: `Jennifer` should provide information from reputable sources and maintain a rigorous process to ensure the quality of information in a consumable and empathetic manner.
- **Extensibility**: `Jennifer` should be easily exten-

---

[*]denotes equal contribution

[1]https://www.newvoicesnasem.org/data-downloads

[2]Facebook http://fb.me/JenniferCOVIDAI

[3]Learn more at: www.NewVoicesNASEM.org

[4]The opinions expressed here are those of the authors and do not represent positions of the National Academies of Sciences, Engineering, and Medicine or the authors' institutions

sible to expand its capability with minimal effort.

The first version of `Jennifer` was released on March 8, 2020. Since then, over 160 volunteers from 141 institutions around the globe recruited through the New Voices' network[5] have helped make daily updates to the chatbot to ensure that its content reflects the latest available information from trusted sources. It is available on the Web and as a Facebook bot. It is also currently embedded in two fact checking systems.[6] As of June 18, 2020, `Jennifer` has been asked 1,480 questions (excluding questions selected via menus and answered 1,059 of them (a response rate of 71%), with an average engagement duration of three minutes and 15 seconds. We plan to conduct more formal evaluation of `Jennifer` in the future.

## 2 Jennifer Overview

We chose to build `Jennifer` as a chatbot, because chatbots "*are able to present concise information from credible sources*" and "*less overwhelming than social media or web search engines' long list of results*" (Miner et al., 2020). The need for agility and speed necessitated that we leverage an existing chatbot platform; rather than building a chatbot from scratch. We chose to utilize Juji (Juji, 2020) that supports both tasks-oriented and social dialogues and allows easy extensions.

This platform supports do-it-yourself chatbot making, similar to Chatfuel[7] and Manychat,[8] but with more advanced NLP capabilities for dialog management similar to Google Dialogflow[9] and IBM Watson Assistant (Janarthanam, 2017; Xiao et al., 2020).[10] By leveraging Juji, we were able to build and deploy the first version of `Jennifer` in less than a day. The resulting chatbot is readily deployable on the Web and as a Facebook bot.

### 2.1 Overall Architecture

Figure 1 depicts the overall architecture of `Jennifer`. As can be seen, `Jennifer` depends on the Juji base system for dialog management (Zhou et al., 2019). Given a user question, Juji uses a pre-trained machine learning model to identify one

---

[5]Learn more about the New Voices Network Tool at https://www.newvoicesnasem.org/the-network

[6]https://coronacheck.eurecom.fr/en and https://idir.uta.edu/covid-19/

[7]https://chatfuel.com

[8]https:manychat.com

[9]https://dialog-flow.com/

[10]https://www.ibm.com/cloud/watson-assistant/

or more relevant question with known answers and depending on its confidence level returns an answer or a follow-up question (more in Sec. 2.2. The main capabilities of `Jennifer` come from the Question-Answer(QA) pairs generated by the extensions specifically implemented for `Jennifer` with two modes of ingestion:

- **Crowdsourced**: This mode relies on a repository of Frequently Asked Questions gathered from reliable sources such as the Centers for Disease Control and Prevention (CDC), the World Health Organization (WHO), the University of Washington Bothell, and the Federation of American Scientists.[11] The questions are provided by the users and volunteers of `Jennifer`, many based on the FAQs. The answers are manually curated by the volunteers of `Jennifer` via a rigorous process detailed shortly.

- **Automated**: Often, users of `Jennifer` ask questions on specific statistics such as number of confirmed cases in a country, or the death rate of a state or a city. The number of questions of this nature was significant, and answers to such questions are changing constantly. Therefore, it is labor intensive to manually curate answers or create alternative questions for such questions. Instead, we have built a QA Generator to automatically create such QA pairs, based on structured data pulled from reliable sources such as the CDC on a daily basis and question templates derived from the crowdsourced questions.

Most QA pairs come from the crowdsourced mode with significant efforts by our volunteers. Our volunteer base is a selected group of medical experts, scientists, engineers, technologists, and specialists. To ensure efficient delineation of tasks and to preserve scientific integrity, we divide this base into four volunteer groups: *Curators*, *Helpers*, *Testers*, and *Admins*, as follows.

Curators take new, unanswered questions, research current answers from reputable and trustworthy sources, and then craft answers with supporting evidence for inclusion. Curators also update answers that have become obsolete. Given the novelty of COVID-19, this is a critical task. Helpers take a set of existing questions and generate many possible question formulations, i.e., alternative questions. This step helps `Jennifer` to better answer unseen questions as Juji fine-tunes its underlying QA engine using additional data. Testers

---

[11]www.cdc.gov, www.who.int, www.uwb.edu, and fas.org

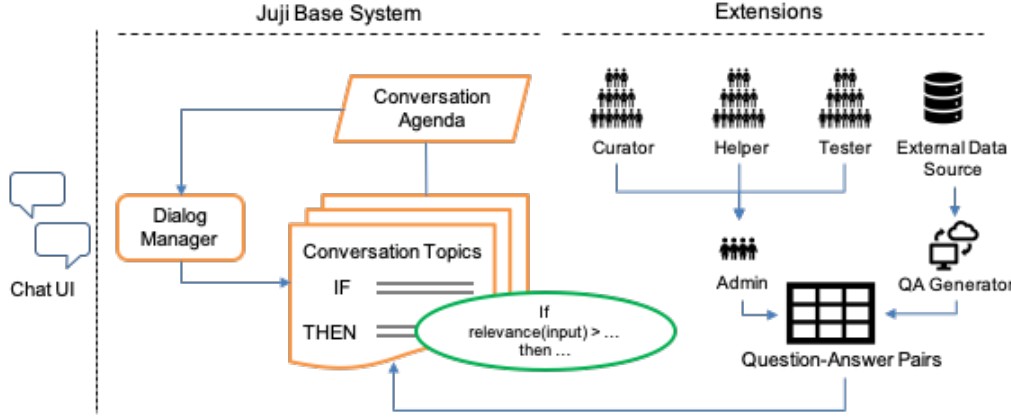

Figure 1: Architecture Overview of Jennifer

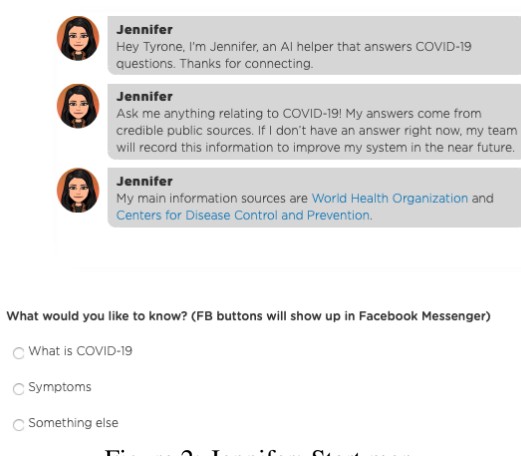

Figure 2: Jennifer: Start menu.

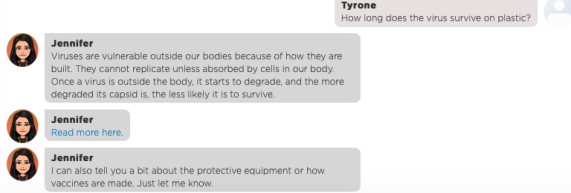

Figure 3: Jennifer informs users about additional topics it knows

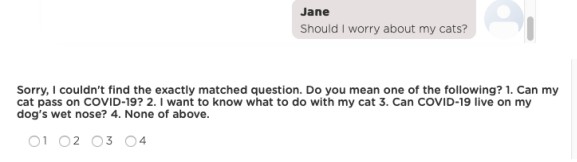

Figure 4: Jennifer recommends relevant questions.

evaluate answers for freshness, accuracy, readability and monitor other possible quality issues with `Jennifer`, e.g., format issue. Input from volunteers is further validated by Admins before it is deployed into `Jennifer`. Specifically, Admins validate all answers, first for scientific validity, and then for language fluency and naturalness in response. Admins also validates alternative questions for relevancy and language fluency. Dedicated Slack channels[12] for each volunteer groups were used to facilitate discussions and collaborations.

## 2.2 Chat Design

We designed the dialog flow of `Jennifer` based on two principles: 1. Fostering *mixed-initiative*

interaction (Walker and Whittaker, 1990); 2. Supporting *Two-way adaptation*: learning from users and also encouraging users to learn what Jennifer can do (Pan et al., 2005)

When `Jennifer` was first launched in early March, most people knew little about COVID-19 or its impact. Thus `Jennifer` started with a "menu" to inform users about its existing knowledge on most important topics (Fig. 2). After answering a question, `Jennifer` also volunteers information on additional topics that it knows (Fig. 3). This design aims to address two challenges: 1) the user may not know how to get started or lack knowledge to ask additional questions; 2) `Jennifer` (or any AI system) will never be perfect; there will always be questions that it cannot answer. By informing users about what it knows, users are more likely to ask questions that `Jennifer` can answer. If `Jennifer` is unsure about how to answer a question, it will recommend similar questions to give users a chance to obtain desired answers as well as learn more about `Jennifer`'s capabilities. Fig. 4 shows how it expresses its uncertainty regarding the user's question but proceeds to recommend a list of relevant inquiries. `Jennifer` will improve its response to similar questions based on user interactions.

`Jennifer` aims at fostering mixed-initiative interactions. On the one hand, it proactively solicits questions from users; on the other hand, it allows users to initiate their questions any time during the chat flow. Such mixed-initiative interactions keep users engaged while allowing users to obtain

---

[12]https://slack.com

information at their own pace.

## 2.3 QA Pairs

As illustrated in Table 1, QA pairs in `Jennifer` are grouped by their ids[13]; questions with the same id are regarded as similar and associated with one or more (semantically equivalent) answers. We release COQB-19 with questions gathered from our users and volunteers.

To be included in the chatbot, each answer needs to satisfy the following criteria:

**Easy to understand**: The information is presented in language understandable by the general public.

**Accuracy and Openness**: The answers must be backed up by data from reliable sources, include references or links to such sources, and be verified by at least one trusted volunteer medical expert. Furthermore, scientific understanding of COVID-19 is quickly evolving; it is important to be explicit about potential uncertainty in the answers.

**Demonstration of Empathy**: The language provided in the answers should emulate natural empathetic conversation, and must acknowledge factors such as stress or anxiety experienced by the users to help foster trust.

## 2.4 Multilingual Support

We have received numerous requests from users around the world to offer `Jennifer`-like capabilities in other languages. On the surface, this task appears to be straightforward. One can translate QAs from `Jennifer` into another language using machine translation (ML). However, language expansion needs to overcome several major obstacles:

- **Language Fluency**: Results produced by commercially available ML services still require significant manual refinement, particularly for domain-specific text (e.g., many answers provided by `Jennifer`).
- **Domain Customization**: Specialized domains such as epidemiology and public health often have their own specific terminologies in non-English languages.
- **Relevancy**: Answers to questions should be verified against reliable sources in the language of question. Additionally, cultural aspects and differences among dialects must be considered when crafting answers in different languages.
- **Models** The Dialog Manager relies on pre-trained machine learning models which usually perform poorer for non-English languages.

---

[13]Manually assigned by Testers and validated by Admins

We therefore chose to expand one language at a time. The first language we selected is Spanish, spoken by 13.5% of the US population.[14] Specifically, we designed and built `Sofia`,[15] also using the Juji platform. The QA collection underlying `Sofia` consists of QA pairs manually translated from the `Jennifer` QA pairs. It is maintained and manually curated by a group of bilingual Spanish-English certified medical interpreters, using verified information from the Spanish language websites of the CDC and WHO. Plans to expand `Jennifer` to other languages are currently under development.

## 3 Discussion

`Jennifer` has successfully demonstrated that, with the right combination of technology and human experts, information from reputable sources can be more quickly and effectively organized and shared at scale. In this section, we share lessons learned and open challenges in the hope to shed some light on promising future research directions.

### 3.1 Lessons Learned

**People are eager to help.** Many scientists and health professionals are eager to step up and help to better respond to the COVID-19 crisis. Many expressed gratefulness for the opportunity to contribute to `Jennifer` as a volunteer.

**Process and Communication is Important.** Given the evolving tasks and the large number of volunteers with diverse background, putting the right process around tasks, workflow, and sequencing (Norheim-Hagtun and Meier, 2010) is key to ensuring efficient use of the volunteers' time to the advantage of the project. It is also important to hold regular dialog with the volunteers to both provide and obtain feedback as well as keep them posted about the progress of the project.

**Effective and Dedicated Management is Critical.** Even with delineation and process optimization, the job of managing volunteers and the intake process requires constant focus and dedication by a few individuals to ensure successful execution. As such, we need to support `Jennifer` with more dedicated resources along with its large number of volunteers to ensure its long-term success.

**Human-Machine Conversation requires Proactive Design.** Despite of the careful chat design

---

[14]https://www.census.gov/data

[15]Available at https://bit.ly/SofíaAI and on Facebook https://fb.me/SofiaCOVIDAI

| ID | Question | Answer |
|---|---|---|
| ChildrenRisk | Are kids at risk? | Based on the current data, nobody seems to be immune from COVID-19, including children. It is true that the number of cases in children is so far lower than the number of cases in adults. We really don't know why this is. The CDC provides answers to commonly asked questions about COVID-19 in here. For those interested in recent research on the subject, a study describing infections in kids in China is available here. |
| ChildrenRisk | Can children be infected? | |
| ChildrenRisk | Are children at risk of getting COVID-19? | |
| ChildrenRisk | Tell me how COVID-19 affects children? | |
| ChildrenRisk | Tell me if kids get infected? | |
| ChildrenRisk | Tell me if children get infected? | |

Table 1: Example QA Pairs

described in Sec. 2.2, improvements on our current design are still desired to avoid the perils of over-promising and encourages users to frame their questions with more specific keywords and simpler sentence structures. We are currently exploring different design options.

## 3.2 Open Challenges

Coordinating the distribution of information at the national level is critical to prepare for the next pandemic (Alexander, 2020). `Jennifer`-like chatbots may be a fundamental component of future misinformation resolution strategy. Our experience with `Jennifer` confirms that it is possible to collaboratively build such chatbots quickly and effectively and to scale these initiatives with the help from many volunteers. However, building these chatbots also comes with its own set of open challenges.

**Scalable Crowdsourced Fact Checking Platform**. Much of the recent research has focused on automating the task of fact checking (e.g., Adair et al. (2017); Pathak and Srihari (2019)). However, in a novel crisis like COVID-19, facts are quickly changing. It is crucial to engage human experts in the loop to ensure the timeliness and accuracy of the answers provided by systems like `Jennifer`. Much of the development and ongoing maintenance of `Jennifer` relies on a rigorous, manual process for quality assurance. Though receiving input from a large number of distributed volunteers is desirable, it remains an open challenge to design, construct, and maintain a fact-checking platform that supports a rigorous process to both engage a large number of experts with diverse expertise levels and leverage automation in minimizing human efforts (Hughes and Tapia, 2015).

**Zero-Shot Empathetic Natural Language Generation (NLG)**. To ensure accuracy, comprehensibility, and appropriate level of empathy, answers provided by `Jennifer` are either manually curated or auto-generated with manually curated templates (Sec. 2). While it is possible to scrape FAQs automatically from reliable resources, how to use the scraped text to generate empathetic answers with little or no training data remains an open problem (Liu et al., 2020), potentially solvable via approaches similar to politeness transfer (Madaan et al., 2020). Identifying multiple resources relevant to a question and composing answers based on them in a coherent and emphathetic manner is an even more challenging problem.

**Competing Information Sources and Public Trust**. Tensions among centralized knowledge networks, such as public health organizations and medical academia, and decentralized information sources on social media platforms and independent news sites introduce new challenges for combating misinformation during global crises. Evidence-based, peer-reviewed information has to compete for public attention and public trust (Cary Funk, 2020). Information literacy becomes ever more important. Solving this challenge requires more than technological innovation (Goldstein, 2020).

## 4 Conclusion

This paper introduces `Jennifer`, a chatbot created to provide easily accessible information to answer questions related to the current COVID-19 pandemic. `Jennifer` leverages cutting-edge chatbot technology, as well as a diverse network of volunteers from around the globe, to combat misinformation related to the pandemic. The information provided by `Jennifer` covers a wide variety of topics, ranging from updated case statistics to food safety and best practices to prevent the virus spread.

## Acknowledgments

The authors would like to acknowledge the National Academies of Sciences, Engineering, and Medicine and the Gordon and Betty Moore Foundation for their generous support of the New Voices project, as well as the guidance and support of the Juji.io team. We would also like to thank our hundred of volunteers whose efforts have made `Jennifer` possible.

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

## A  Auto-Generation of QA Pairs

Below is part of the data from CDC website on June 30, 2020. Jun 30 2020 12:15PM Case and Death data . Testing data updated as of Jun 25 2020 12:00AM

For statistics-related questions, we apply a simple template-based approach to automatically generate the corresponding QA pairs so that we can update `Jennifer` with the most current information while minimizing manual efforts. The templates were all manually curated by Admins based on input from other volunteers.

Specifically, we current generate the following statistics-related questions automatically and deploy them on regular basis:

- Number of confirmed cases in a specific U.S. state/jurisdiction using Case and Death data from CDC.

- Number of death in a specific U.S. state/jurisdictions using Case and Death data from CDC.

- Number of confirmed cases in a country using Case data from WHO.

Table 2 shows the example templates for QA pairs within a single group. Based on data released by CDC on daily basis (e.g. Table 3), we automatically generate QA pairs from the templates, as illustrate in Table 4.

## B  COQB-19: COVID-19 Question Bank

This dataset is derived from `Jennifer`, a chatbot designed and built to provide information on COVID-19 from trusted resources (available at `http://bit.ly/jenniferai` and on `http://fb.me/JenniferCOVIDAI`). The dataset is intended to enable researchers to understand common questions asked by general public on COVID-19 and to apply recent advances in natural language processing to better answer such questions. The corpus will be updated periodically.

### B.1  Description

The current dataset consists of a total of 3,924 questions. Each question is assigned with a *Question ID*. Questions of the same ID are regarded as similar questions and grouped together in the dataset.

Based on the original source of the questions, the dataset is further divided into two subsets:

- **COQB-19**$_{crowdsourced}$ This set consists of a total of 2,341 questions with 280 unique Question IDs. They are gathered based on input from users and volunteers of `Jennifer`

- **COQB-19**$_{auto-generated}$ This set consists of a total of 1,583 questions with 664 unique Question IDs. unique Question IDs. They are automatically generated with pre-defined question templates that were designed based on input from users and volunteers of `Jennifer`.

The dataset is available at `http://www.NewVoicesNAMSEM.org/data-downloads` with the Open Covid Pledge OCL-PC v1.1 license ( `https://opencovidpledge.org/licenses/v1-1-ocl-pc/`).

## C  Preliminary Plan for Evaluation

We plan to conduct comprehensive empirical studies to evaluate `Jennifer` in the near future.

First of all, we are interested in evaluating the usefulness and effectiveness of `Jennifer` from the end users' perspective. Specifically, we plan to investigate the following questions.

- Are the answers provided by `Jennifer` easy to understand?

- Are the answers provided by `Jennifer` relevant to the users' questions?

- Are the answers provided by `Jennifer` useful to the users?

- How do the users find the interaction with `Jennifer`?

- How successful is `Jennifer` in fostering trust with the users?

We seek to answer these questions via user studies by questionnaires as well as by measuring overall user behavior when interacting with the system, such as activation rate, engagement duration, volunteer user engagement, and confusion rate.

Secondly, we are interested in evaluating how QA pairs of `Jennifer` evolve over time and measures the effectiveness of the overall flow. Specific, we plan to investigate the following questions.

- Are the answers provided by Curators of high quality?

| ID | Questions | Answers |
|---|---|---|
| Case⟨STATENAME⟩ | How many confirmed cases in ⟨STATENAME⟩? | There were ⟨NUMCASE⟩ cases |
| Case⟨STATENAME⟩ | How many cases have been confirmed in ⟨STATENAME⟩? | reported in ⟨STATENAME⟩ |
| Case⟨STATENAME⟩ | How many people in ⟨STATENAME⟩ have it? | of ⟨DATE⟩. However, the |
| Case⟨STATENAME⟩ | Number of confirmed cases of COVID 19 in ⟨STATENAME⟩? | numbers are changing every day. |
| Case⟨STATENAME⟩ | How many COVID-19 cases have been confirmed in ⟨STATENAME⟩? | For regular updates, please go to |
| Case⟨STATENAME⟩ | How many people have been tested positive for COVID-19 in ⟨STATENAME⟩? | CDC website (⟨URL⟩). |
| Case⟨STATENAME⟩ | How many confirmed cases are there in ⟨STATENAME⟩? | |

Table 2: Sample QA templates

| abbr | fips | jurisdiction | Total Cases | Total Death | Death100k | CasesInLast7Days | RatePer100000 |
|---|---|---|---|---|---|---|---|
| AK | 2 | Alaska | 904 | 14 | 1.9 | 149 | 122.6 |
| AL | 1 | Alabama | 37203 | 931 | 19 | 7182 | 761.1 |
| AR | 5 | Arkansas | 20257 | 265 | 8.8 | 4696 | 672.1 |

Table 3: Partial COVID-19 related Case and Death data updated as of Jun 30 2020 12:15PM on CDC website.

- Are the questions provided by Helpers of high quality?

- Are the feedback provided by Testers of high quality?

- How much time on average each QA pair update takes (for the crowed-sourced questions)?

- How does `Jennifer` evolves over time in terms of volunteer efforts?

- How effective is the current process used by `Jennifer`

We have daily logs of changes to the QA pairs to track input from the volunteers. By analyzing these logs, we can quantitatively estimate the qualities of volunteers' input (e.g. based on the acceptance of volunteer input by the Admins). In addition, we plan to summarize feedback gathered from Testers to analyzes how `Jennifer` evolves over the past months. We also intend to conduct user studies in a controlled setting to evaluate the current processing as well as sending questionnaires to our volunteers to gather their feedback to evaluate different stages of the existing process.

| ID | Questions | Answers |
|---|---|---|
| CaseAlaska | How many confirmed cases in Alaska? | There were 904 |
| CaseAlaska | How many cases have been confirmed in Alaska? | reported in Alaska |
| CaseAlaska | How many people in Alaska have it? | of June 30, 2020. However, the |
| CaseAlaska | Number of confirmed cases of COVID 19 in Alaska? | numbers are changing every day. |
| CaseAlaska | How many COVID-19 cases have been confirmed in Alaska? | For regular updates, please go to |
| CaseAlaska | How many people have been tested positive for COVID-19 in Alaska? | CDC website (`https://www.cdc.gov/...`). |
| CaseAlaska | How many confirmed cases are there in Alaska? | |
| CaseAlabama | How many confirmed cases in Alabama? | There were 37203 |
| CaseAlabama | How many cases have been confirmed in Alabama? | reported in Alabama |
| CaseAlabama | How many people in Alabama have it? | of June 30, 2020. However, the |
| CaseAlabama | Number of confirmed cases of COVID 19 in Alabama? | numbers are changing every day. |
| CaseAlabama | How many COVID-19 cases have been confirmed in Alabama? | For regular updates, please go to |
| CaseAlabama | How many people have been tested positive for COVID-19 in Alabama? | CDC website (`https://www.cdc.gov/...`). |
| CaseAlabama | How many confirmed cases are there in Alabama? | |
| CaseArkansas | How many confirmed cases in Arkansas? | There were 20257 |
| CaseArkansas | How many cases have been confirmed in Arkansas? | reported in Arkansas |
| CaseArkansas | How many people in Arkansas have it? | of June 30, 2020. However, the |
| CaseArkansas | Number of confirmed cases of COVID 19 in Arkansas? | numbers are changing every day. |
| CaseArkansas | How many COVID-19 cases have been confirmed in Arkansas? | For regular updates, please go to |
| CaseArkansas | How many people have been tested positive for COVID-19 in Arkansas? | CDC website (`https://www.cdc.gov/...`). |
| CaseArkansas | How many confirmed cases are there in Arkansas? | |

Table 4: Sample QA pairs generated using templates in Table 2 from data in Table 3