# OpenReview forum: "Jennifer for COVID-19: An NLP-Powered Chatbot Built for the People and by the People to Combat Misinformation"
_aclweb.org/ACL/2020/Workshop/NLP-COVID — NLP-COVID-2020 Abstractonly_

### Official Review · AnonReviewer3 · 2020-06-26
**N/A**

**Rating:** 7
**Confidence:** 3

**Review:**

The paper provides a good description of the COVID-19 Q&A chatbot and the process of creating the crowd-sourced Q/A pairs that the bot uses for question answering. The authors also outline the opportunities and challenges of implementing COVID-19 chatbots in the real world.
While the study authors list examples of lessons learned as well as open challenges of using chatbots in a pandemic setting in which information changes quickly, it would be very helpful to better understand the use of a Jennifer-like chatbot in a situation in which the evidence-base is more stable. It would also be very helpful to understand the authors' perspectives on how chatbot technologies and their use could be effectively evaluated, and how such an evaluation could be better integrated into the chatbot development and validation of the provided responses. In addition, it might be helpful to provide examples of other chatbot technologies that have been used successfully during and prior to the pandemic, as this could put better into perspective the issues and imitations that are being raised in the manuscript.
Another important aspect regarding the use of chatbot technologies is their acceptance both by public health authorities as well as the intended users. In using novel technologies, governments and public health organizations typically tend to take a more conservative approach, and accept them only following rigorous testing and evaluation. Equally, there may be also hesitancy from certain population groups with formed/strong opinions to engage with chatbots, particularly when provided answers originate from just one or several sources (i.e. when the whole body of evidence is not provided). It remains unclear who is (are) the intended target audience(s) of the chatbot and, ultimately, its objective. It would be very helpful to have that information stated and discussed up front. Lastly, it is important to recognize the significant limitations of the use of chatbot technologies for "fact-checking" in the absence of a robust knowledge-base. What has been presented in the paper may be more applicable to situation in which there is a need to help users identify and navigate through the relevant content, particularly when information is not well structured or ambiguous. It would also be great if the authors could reflect in the sustainability of initiatives that have a very high reliance on volunteers
I would recommend that the manuscript be accepted, as it provides a good example on the potential usefulness of chatbot technologies in supporting the general public to efficiently find credible information on the web.

---

> ### Author Response · Authors · 2020-07-01
> **Thank You and Some Clarifications**
>
> Thank you very much for your insightful comments. We completely agree that Jennifer is a great example to demonstrate the potential usefulness of chatbot technologies in supporting the general public to efficiently find credible information on the web.
>
> >> Better understand the use of a Jennifer-like chatbot in a situation in which the evidence-base is more stable.
> >> Examples of other chatbot technologies that have been used successfully during and prior to the pandemic
>
> To the best of our knowledge, , the currently available literature does not report examples of other chatbots that were designed with a similar scope to Jennifer (Miner et al. 2020, Larajano et al., 2018). We do agree that there is a need to better understand the use of a Jennifer-like chatbot in a situation with more stable evidence-base.  There are several studies conducted before that compare Juji against other chatbot technologies. (https://juji.io/publications; we cited the most recent one Xiao etc. 2020).  We intend to conduct further study when information for COVID-19 is more stable and comapare different chatbots surfaced to help fight against COVID-19. We welcome researchers interesting in the problem reaching out to us for potential collaboration,.
>
> Laranjo L, Dunn AG, Tong HL, et al. Conversational agents in healthcare: a systematic review. J Am Med Inform Assoc. 2018;25(9):1248-1258.
> Miner, A.S., Laranjo, L. & Kocaballi, A.B. Chatbots in the fight against the COVID-19 pandemic. npj Digit. Med. 3, 65 (2020).
>
> >> How chatbot technologies and their use could be effectively evaluated, and how such an evaluation could be better integrated into the chatbot development and validation of the provided responses.
>
> Evaluation of chatbots technologies is an important topic.
>
> The design of Jennifer includes three different perspectives, which can be evaluated separately:
> (1) the technical design of Jennifer
> (2) the interaction of human users with Jennifer
> (3) the COVID-19 related information provided by Jennifer
>
> Our core techical design considerations are 1) Rapid development; 2) Ease of access; 3) Ease of maintenance; 4) Quality assurance. Because of (1) and (3), we selected Juji as our platform and was able to build the first version of Jennifer in less than a day. Several studies were conducted before that compare Juji against other chatbot technologies. (https://juji.io/publications; we cited the most recent one Xiao etc. 2020). It would be possible(but take some time) to compare the design of the chatbots reviewed by Larajano et al. 2018 with that chosen for Jennifer. The scope of these chatbots was different, however, as they were designed as assistants for self-diagnosis.
>
> It would also be of interest to evaluation of the acceptance of Jennifer by human users in real life situations. We hope to conduct formal user studies in the future.
>
> Currently, our evaluation and updates to the answers provided by Jennifer are mostly manual with the help from our volunteers. It would be interesting to create more automated evaluation methods and allow users of the chatbot to provide feedback more directly and incorporate them into the chatbot. We plan to explore this further in the future.
> >> Acceptance of chatbot technologies.
>
> Chatbot technologies have been adapted by such organizations to help fight COVID-19. For instance, both WHO and CDC have used chatbot technology for assisting people in self-diagnosis (Miner et al., 2020.) We agree that it is important to provide evidence when giving answers. This is precisely when when Jennifer provides an answer to each question, references/links to the corresponding reliable resources are provided to help foster trust.
>
> >> Intended target audience(s) of the chatbot and, ultimately, its objective
>
> The intended audience is general public who has difficulty navigating information about COVID-19 distributed across different resources.
>
> >> it is important to recognize the significant limitations of the use of chatbot technologies for "fact-checking" in the absence of a robust knowledge-base.
>
> As with other scientific disciplines, knowledge on COVID-19 is evolving and,hence,the answers given by Jennifer have to be regularly reviewed and updated. A chatbot like Jennifer would likely be no good solution to navigate through information that is not well structured or ambiguous because it relies on a human team that performs a careful review ofthe questions that it can understand and the answers that it can give.
>
> >> Sustainability of initiatives that have a very high reliance on volunteers }}
>
> As shared in Sec. 3, we found that we need to support the initiative with more dedicated resources and better infrastructure to manage the large number of volunteers. In general, people are eager to help, but we need to have better support to ensure that their time and efforts are used in the most effective way.

---

### Official Review · AnonReviewer2 · 2020-06-29
**Impressive system, lacking crucial details**

**Rating:** 7
**Confidence:** 4

**Review:**

The authors describe a chatbot and a dataset of frequently asked questions regarding the SARS-CoV-2 disease. They enumerate the challenges in building an effective health chatbot. They introduce a dataset of SARS-CoV2 and COVID19 questions.

I would like to congratulate the authors on having an actual deployed system. I have many concerns about the paper and the datasets themselves that hopefully the authors can address.

Pros:
1. The authors collect what might be a high quality dataset harvested from many QA sources
2. They provide compelling examples of good bot behavior.
3. They provide a reasonable roadmap forward.
4. They are displaying great care in deployment of their system, particularly beyond their initial input domain.
5. I am a huge fan of the ongoing curation of the answers in their dataset.

Concerns:
1. The authors should provide a brief overview of the capabilities from Juji: Does the question/QA grouping come from Juji itself, or do the authors have some other approach? (From a brief skim of the Juji paper it seems that the authors must have taken some alternative approach)
2. There is no measure of question/QA grouping. Presumably this should be a very high precision grouping since incorrectly including a question could mean it has a wrong answer.
3. Similarly, there is no measure of inter-rater agreement on the evaluation of the answers in the crowd sourced context. A measure such as Krippendorf's alpha (or many others) would be appropriate here and essential to knowing how well the system performs overall. A measure like this becomes doubly important when the authors are attempting to fight the spread of misinformation.
4. How were the automatically generated question generated? If you used Juji templates, what are some examples? Given the constraints of available structured data (e.g. CDC data), what types of questions did you choose to answer, and which ones did you not?
5. For the questions gathered from multiple sources, how are conflicts resolved? Presumably two sources do not always have consistent answers.
6. How many people evaluate each question/answer pair?
7. How is the translation performed? I can't tell if it's automatic with manual curation or entirely manual. If there was an automatic component, what system did this?
8. I am concerned about the use of "engagement" as a metric of success. This becomes even more concerning to me when more time spent interacting with a system is seen as a measure of its success - while I agree that when answering questions one might need further clarification or to know about related questions and answers - encouraging people to spend more time interacting with a system could lead to a perverse incentive where it keeps users "engaged" without answering their questions.

The authors have built an impressive system, but I am reluctant to recommend it for acceptance without further details in data generation and evaluation. If these details were added (even for the system/questions/answers at a particular point in time), I would gladly recommend this paper for acceptance.

---

> ### Author Response · Authors · 2020-07-01
> **Response to the Concerns Raised**
>
> Thank you for your detailed comments and feedback. We have seemed to address all concerns in the revised version as detailed below.
>
> >> The authors should provide a brief overview of the capabilities from Juji
> The main capabilities Juji provides is Dialog Management, including given a user question, matching the question to a known question based on a pre-trained machine learning model and return the corresponding answer for the question group where the question below to. The ML model also gradually fine-tuned to the specific workspace based on user feedback (e.g. selection on ambiguous questions).  We have added a brief description about to the beginning of Sec. 2.1 (this is a 4-page short paper submission, hence the brevity).
>
> >> 2. There is no measure of question/QA grouping. Presumably this should be a very high precision grouping since incorrectly including a question could mean it has a wrong answer.
>
> The grouping is done manually, so it is very precise. We've clarified this at the beginning of Sec. 2.3.
>
> >> 3. Similarly, there is no measure of inter-rater agreement on the evaluation of the answers in the crowd sourced context.
>
> The measure of inter-annotator agreement doesn't apply, as we only add/update an answer when it passes both validation stages by trusted medical experts. Similarly an alternative question is added only when it passes validation by trusted admins.  We've clarified this further towards the end of the last paragraph of Sec. 2.1.
>
> >> 4. How were the automatically generated question generated? If you used Juji templates, what are some examples? Given the constraints of available structured data (e.g. CDC data), what types of questions did you choose to answer, and which ones did you not?
>
> We have provided more details in Appendix A. To sumamrized, we focus on answer questions related to case numbers and death rates in US states with data from CDC and different countries with data from WHO. The auto-generated questions are also included in the COQB-19 dataset. We also provided more details in Appendix B about the data.
>
> >> 5. For the questions gathered from multiple sources, how are conflicts resolved? Presumably two sources do not always have consistent answers.
>
>
> The answers are manually curated by volunteers and validated by trusted medicine experts. When two resources are conflicting with each other, we will provide an answer based on the most authoritative resource (e.g. CDC) but also include the alternative answer as appropriate. We rely on our trusted volunteer medicine experts to make the judgement call.
>
> >> 6. How many people evaluate each question/answer pair?
> The exact number varies. In general, every answer has been evaluated by at least three volunteers: one curator, one tester, and one admin. As information related to COVID-19 is evolving over time (e.g. recommendation on the usage of masks), some of the answers were revised, tested, and validated more than once.
>
> >> 7. How is the translation performed? I can't tell if it's automatic with manual curation or entirely manual. If there was an automatic component, what system did this?
>
> It is done manually. As questions and answers are added to the Jennifer QA database, our volunteer translators incorporate the respective QA translations and links to trusted sources. We've clarified this point in the last paragraph of Sec. 2.4.
>
> >> 8. I am concerned about the use of "engagement" as a metric of success. This becomes even more concerning to me when more time spent interacting with a system is seen as a measure of its success - while I agree that when answering questions one might need further clarification or to know about related questions and answers - encouraging people to spend more time interacting with a system could lead to a perverse incentive where it keeps users "engaged" without answering their questions.
> We agree that that "engagement" should not be the ultimate measure of success of our system. We do plan to conduct formal user studies in the future to evaluate Jennifer with different metrics such as NPS and user sentiment.

---

> > ### Comment · AnonReviewer2 · 2020-07-03
> > **Going the right direction!**
> >
> > Overall, I'm pretty pleased with the updates and will bump my score a little bit; remaining points below. Basically I want a plan for evaluation before I become enthusiastic about accepting this.
> >
> > > The measure of inter-annotator agreement doesn't apply, as we only add/update an answer when it passes both validation stages by trusted medical experts. Similarly an alternative question is added only when it passes validation by trusted admins. We've clarified this further towards the end of the last paragraph of Sec. 2.1
> >
> > a) each of these stages is still probabilistic, you already understand this as you have multiple stages of QA
> > b) if something can be rejected at any stage, you do in fact have something to measure agreement against (it's not a perfect measure, but it would be better than nothing)
> > c) if you don't have some measure of agreement, you should at least have a plan for collecting one
> >
> > Having something like this, in mind, becomes effectively non-negotiable for full conference publications, but as a work-in-progress I care more about a long-term plan.
> >
> > >      How many people evaluate each question/answer pair? The exact number varies. In general, every answer has been evaluated by at least three volunteers: one curator, one tester, and one admin. As information related to COVID-19 is evolving over time (e.g. recommendation on the usage of masks), some of the answers were revised, tested, and validated more than once.
> >
> > A less important note - if you track sources and different versions of questions/answers, this could be a valuable resource for investigating the flow/change of information in a relatively fast moving world.
> >
> >
> > > We agree that that "engagement" should not be the ultimate measure of success of our system. We do plan to conduct formal user studies in the future to evaluate Jennifer with different metrics such as NPS and user sentiment.
> >
> > This should probably be detailed, at least in an appendix

---

> > > ### Author Response · Authors · 2020-07-03
> > > **Preliminary Plan for Evaluation Added**
> > >
> > > Thank you for your positive feedback and insightful comments.
> > >
> > > We have added Appendix C. Preliminary Plan for Evaluation with initial thoughts on how we plan to evaluate Jennifer from both users' as well as volunteers' perspective and both quantitively as well as qualitatively. Please do let us know if you have further comments or suggestions.
> > >
> > > We also plan to seek collaborations with experts in human-computer interaction to carry out some of the studies. We welcome any researcher interested in such opportunities to contact us directly.

---

### Official Review · AnonReviewer1 · 2020-06-29
**Engineering white-paper, but regrettably lacking in science**

**Rating:** 4
**Confidence:** 4

**Review:**

Summary:
the article describes the process and end result of creating a chatbot aimed at providing vetted COVID-related information to the public.

General impression:
Jennifer clearly represents a considerable effort, undertaken by a talented group under great time pressure, using vast amounts of information that is highly uncertain and in flux.

Now for the submission: to put it perhaps soberingly, it is 60% helicopter-view engineering white paper, 20% sales pitch, 10% incidental observations, 8% future work, and 2% science. It is the virtual absence of scientific substance that is my main concern.
As one illustration: the manuscript has 24 citations, 11 of those look peer-reviewed, 5 of these are provided in the future work (‘ongoing challenges’) section, therefore not supporting the past activities but rather motivating future ones. Several of the other ones are referred to outside of their scope (e.g. the Vargas reference). Related research on chatbox design, the spine of this work, is all but absent. (2 citations, from 30 and 15 years ago).

Note that the submission has no ‘Results’ section. No systematically collected observations or other evaluation materials are introduced. At the very least, end-user usage statistics would have been informative to assess the impact of this work.
The engineering decisions that are reported make sense, but are largely not substantiated with supporting literature or first-hand evaluation.

While it is claimed, it is not clearly demonstrated that this project addresses misinformation (or disinformation). It is an access point (and repeater of) trustworthy information but not one that specifically points out misinformation, or hunts it down. It is not fully clear how accountability is ensured for the automated provision of information to the general public – can users easily vet the sources of such information, and is there a path of recourse when a user spots incorrect answers? Who is responsible for the answer in such a case, especially so if it led to harm?  Are there mechanisms to prevent deliberate infiltration of the expert-crowd or other efforts to sabotage Jennifer (think of Tay AI)?

More detailed points:
The Abstract refers to COQB, but the main article never does.

Section 2.4: "Answers to questions should be verified against reliable sources in the language of question."
--> That is a debatable statement: why would an English language source suddenly be necessarily inadequate when a person asks a question in French!?

Discussion: "Jennifer has successfully demonstrated that, with the right combination of technology and human experts, information from reputable sources can be more quickly and effectively organized and shared at scale."
--> This is a statement that is not supported by anything reported in the submission.

"Given the evolving tasks and the large number of volunteers with diverse background, putting the right process around tasks, workflow, and sequencing (Norheim-Hagtun and Meier, 2010) is key to ensuring efficient use of the volunteers’ time to the advantage of the project."
--> Unfortunately, full documentation of tasks, workflow, and sequencing is not provided in this manuscript, and use of volunteer time is not assessed.

Conclusions:
"Jennifer leverages cutting-edge chatbot technology, as well as a diverse network of volunteers from around the globe, to combat misinformation related to the pandemic."
--> Only one of these three claims is supported by the presented material.

At best, this submission could be included as a discussion article, certainly not a research article. Accepting it as a full peer-reviewed research contribution would lend it more credibility than it deserves.

Minor points, stylistic points:
"to best prevention and management practices." -->?? "to prevention and management best-practices."

Section 2.3: " following guidelines and recommendations in (National Academies of Sciences, Engineering, and Medicine, 2017)". The citation appears to point to a research agenda document, rather than a set of guidelines and recommendations.

The sentence "Much of the recent research focusing on automating the task of fact checking (e.g., Adair et al. (2017); Pathak and Srihari (2019))." is incomplete.

Several citations are incomplete re. sources, most notably the Jurkowitz (Mark, not Marl) citation.

---

> ### Author Response · Authors · 2020-07-01
> **Response to the Concerns Raised.**
>
> We thank the reviewer for the helpful comments.
>
> At the onset of COVID-19, our main goal was not so much creating novel scientific knowledge but quickly and effectively disseminating validated information about COVID-19 to help people to better protect themselves against this disease. To race against time, we chose to rely on existing technologies that may not be perfect and add scientific evaluation later during the field work. We believe that sharing our experience of building Jennifer help shred lights on research directions to help fight the current and future crisis. We also release a dataset that can be used to advance NLP research (e.g. QA and semantic similarity).
>
> >> Concerns about citations.
>
> References are provided to support information and statements incorporated in the article. We have added more citations to recent work on chatbots and converted some to footnotes.
>
> >> the submission has no ‘Results’ section.
>
> The usage statistics are provided in Sec. 1. We do intend to conduct formal user studies in the future.
>
> >> The engineering decisions that ... largely not substantiated with supporting literature or first-hand evaluation.
>
> Jennifer is a proof-of-principle system. This paper describes the first steps in its design, implementation, and initial evaluation.  The chat design is informed by existing literature (e.g. Walker and Whittaker, 1990 and Pan, et al. 2005). To the best of our knowledge, the currently available literature does not report examples of other chatbots that were designed with a similar scope to Jennifer (Miner et al. 2020, Larajano et al., 2018). We do intend to conduct formal user studies in the future.
> >> it is not clearly demonstrated that this project addresses misinformation
>
> Social media platforms and several websites available to the general public are plagued with misinformation. The UN secretary-general has warned we’re living through “a pandemic of misinformation,” and the head of the WHO has said we are in an “infodemic.” By being easily accessible to the general public, this platform combats misinformation by redirecting users to trusted sources, and answering questions in a way that satisfies the criteria described in section 2.3.
>
> In addition, the WHO has published a COVID-19 myth busters page with questions such as whether the virus would be a bioweapon. Jennifer provides answers for such questions as well.
>
> >> It is not fully clear how accountability is ensured
>
> The answers are manually curated by volunteers and validated by trusted medicine experts. Answers always include a link to a trusted source, so that end users can "learn more". In general, every answer is evaluated by at least three volunteers: one curator, one tester, and one administrator. As information related to COVID-19 is evolving over time (e.g. the usage of masks), some answers were revised, tested, and validated more than once. We have a dedicated team of volunteers who revise QA pairs as information evolves. In addition, when resources are conflicting, we provide an answer based on the most authoritative resource (e.g. CDC) but also include the alternative answer as appropriate. We rely on our trusted volunteer medicine experts to make the judgement call.
>
> >> The Abstract refers to COQB, but the main article never does.
>
> We've added a reference to the data to the beginning of Sec. 2.3 and more details to Appendix B.
>
> >> why would an English language source suddenly be necessarily inadequate when a person asks a question in French!?
>
> While we agree that the main scientific concepts in an answer do not change with language, there are language style and cultural differences that need to be considered when drafting an answer. We also want to make sure that relevant information is available in language-specific websites, and direct end-users to trusted sources. Furthermore, we do not believe that we could assume that our readers understand English.
>
> >>  "Jennifer has successfully demonstrated that, ...."
> --> This is a statement that is not supported by anything reported in the submission.
>
> As we discuss in the text, Jennifer facilitates access to information by providing easy to understand, empathetic, and curated answers to end-user questions, while redirecting them to reputable sources. Rather than browsing independently and potentially accessing unreliable information, end users can rely on Jennifer to provide a trustworthy answer, that is backed up by reliable sources.
>
> >> full documentation of tasks, workflow, and sequencing is not provided in this manuscript, and use of volunteer time is not assessed.
>
> Sec. 2.1 describes the workflow and sequencing of tasks. Some details are omitted due to space limit.
>
> >> Only one of these three claims (in conclusion) is supported by the presented material.
>
> We have modified the text to emphasize areas where these claims are supported.
>
> We have fixed all other minor issues related to citation and writing as pointed by the reviewer.

---

### Decision · Program_Chairs · 2020-07-04

**Decision:**

Accept (Abstract only)

**Comment:**

This paper introduces a system which leverages a chatbot as a tool for public dissemination of health information related to COVID-19, and describes the use of human experts to craft responses to questions.

However, the paper itself is somewhat ambiguous about the specific role of the chatbot and the NLP aspects of the system in "combatting misinformation". Specifically although there is significant work in the NLP community to leverage automated analysis methods for fact checking and flagging incorrect information, that is not what the system does -- ultimately it is the human curators and admins that do all of the combatting of misinformation, while the chat bot is simply the means for accessing and disseminating that information. From an NLP perspective, the contributions of the work are primarily the QA pairs that are produced (the data set), although we are not told specifically how those pairs are used, i.e. how are user questions mapped into questions in the Question bank (beyond "The  Dialog  Manager relies on pre-trained machine learning models" -- what models? relies how? how do they work?).

From a broader application perspective, as the reviewers have identified, a nice example of how NLP can be combined with a public health activity, and points to the value of chat bots for supporting dissemination of information to the public, but there are unaddressed questions about the actual quality of the information that results from this process, and the acceptability of this sort of approach to public health officials.

In short, the system represents a potentially valuable public health contribution based on NLP and a nice example of crowd-sourcing, but the presentation in this manuscript does not sufficiently address the role of NLP in achieving the stated objectives.

We nevertheless think it is interesting to hear about this work in the context of the workshop and would therefore like to invite you to present your system at the workshop (10 minute presentation), and to provide a 1-page summary of the system and the data set for inclusion in the post-proceedings; instructions will follow after the workshop.